# A Simple Calibration Method to Consider Plastic Deformation Influence on X-ray Elastic Constant Based on Peak Width Variation

Ewann Gautier [1], Pierre Faucheux [1,2], Bruno Levieil [1,*], Laurent Barrallier [2], Sylvain Calloch [1] and Cédric Doudard [1]

[1] ENSTA Bretagne, IRDL—UMR CNRS 6027, 2 Rue François Verny, 29200 Brest, France; ewann.gautier@ensta-bretagne.org (E.G.); sylvain.calloch@ensta-bretagne.org (S.C.)

[2] MSMP Arts et Métiers Institute of Technology, HESAM Université, F-13617 Aix-en-Provence, France; laurent.barrallier@ensam.eu

[*] Correspondence: bruno.levieil@ensta-bretagne.org

**Abstract:** The $\sin^2 \psi$ method is the general method for analyzing X-ray diffraction stress measurements. This method relies on the estimation of a parameter known as $\frac{1}{2}S_2^{hkl}$, which is generally considered as a material constant. However, various studies have shown that this parameter can be affected by plastic deformation leading to proportional uncertainties in the estimation of stresses. In this paper, in situ X-ray diffraction measurements are performed during a tensile test with unloads on a low-carbon high-strength steel. The calibrated $\frac{1}{2}S_2^{hkl}$ parameter varies from $3.5 \times 10^{-6}$ MPa$^{-1}$ to $5.5 \times 10^{-6}$ Mpa$^{-1}$, depending on the surface condition and on the plastic strain state, leading to a maximum error on the stress level of 40% compared to reference handbook values. The results also show that plastic strain is responsible for 6 to 14% of the variation, depending on the initial surface sample condition. A method is then proposed to correct this variation based on the fit of the $\frac{1}{2}S_2^{hkl}$ evolution with respect to the peak diffraction width, the latter being an indication of the plasticity state. It is shown that the proposed methodology improves the applied stress increment prediction, although the absolute stress value still depends on pseudo-macrostresses that also vary with plastic strain.

**Keywords:** X-ray elastic constants; plastic strain; pseudo-macrostress

## 1. Introduction

Accurately estimating the residual stress state is essential for sizing structural parts, as stresses affect the mechanical behavior, fracture resistance, and fatigue life. Residual stresses can be characterized experimentally, for example, with diffractometric methods, such as X-ray diffraction and neutron diffraction, or with mechanical methods, such as hole drilling and slitting [1]; or numerically, using simulations of manufacturing processes [2,3]. It should be noted that manufacturing process simulations also depend on experimental residual stress measurements for validating the models.

This work focuses on residual stress measured using laboratory X-ray sources and the $\sin^2 \psi$ method, which is the de facto standard method for measuring residual stresses via X-ray diffraction in materials with little or no texture and grains that are small compared to the X-ray irradiated volume (see, for example, standards [4–6], among others). The $\sin^2 \psi$ method consists in measuring the diffracted angle 2Θ for different inclinations of the incident X-ray beam with respect to the surface of the specimen, estimating the average lattice strain in the direction normal to the diffraction vector in the diffracting regions using Braggs law, and then computing stresses using Equation (1):

$$\begin{aligned}
\varepsilon_{\varphi\psi}^{hkl} = {} & \tfrac{1}{2}S_2^{hkl}\left[\sigma_{11}\cos^2\varphi + \sigma_{12}\sin(2\varphi) + \sigma_{22}\sin^2\varphi - \sigma_{33}\right]\sin^2\psi \\
& + \tfrac{1}{2}S_2^{hkl}\left[\sigma_{13}\cos\varphi + \sigma_{23}\sin\varphi\right]\sin(2\psi) \\
& + S_1^{hkl}\left[\sigma_{11} + \sigma_{22} + \sigma_{33}\right] + \tfrac{1}{2}S_2^{hkl}\sigma_{33}
\end{aligned} \tag{1}$$

This equation relates average lattice strains $\varepsilon_{\varphi\psi}^{hkl}$ measured in the $(\varphi,\ \psi)$ direction to the components $\sigma_{ij}$ of the macroscopic stress field present in the material. In Equation (1), directions 1 to 3 are as defined in Figure 3, $\varphi$ and $\psi$ are 'sampling angles' that characterize the orientation of the diffraction vector with respect to the normal to the surface of the specimen [7], and $S_1^{hkl}$ and $S_2^{hkl}$ are material constants that depend on the lattice planes $\{hkl\}$ being observed, called X-ray elastic constants (XECs). This model assumes that the material is made of crystallites that are small compared to the X-ray irradiated volume, and that crystallites are oriented perfectly randomly—i.e., that the material has no texture. For a detailed derivation of the model, see, for example, [8]. It should be noted that a model is needed to compute stresses from $\varepsilon_{\varphi\psi}^{hkl}$ because X-ray diffraction only provides information about (i) average strains (ii) in the direction normal to the lattice planes being observed (iii) in diffracting crystallites only (that represent a small subset of all crystallites), and that, unless the material is made of perfectly elastically isotropic crystallites, different subsets of crystallites will deform differently.

If measurements are performed at a constant rotation angle, say $\varphi = 0°$, Equation (1) is simplified to

$$\varepsilon_{\varphi=0°,\ \psi}^{hkl} = \frac{1}{2}S_2^{hkl}\sigma_\varphi\sin^2\psi + \frac{1}{2}S_2^{hkl}\tau_\varphi\sin(2\psi) + C, \tag{2}$$

with

$$\sigma_\varphi = \sigma_{11} - \sigma_{33},\ \ \tau_\varphi = \sigma_{13},\ \ C = S_1^{hkl}[\sigma_{11} + \sigma_{22} + \sigma_{33}] + \frac{1}{2}S_2^{hkl}\sigma_{33}, \tag{3}$$

where the unknowns $\sigma_\varphi$, $\tau_\varphi$, and $C$ can be determined by fitting Equation (2) to several $\varepsilon_{\varphi\psi}^{hkl}$ measured at different $\psi$ angles.

On the other hand, for quick or quality control measurements, the X-ray elastic constants $S_1^{hkl}$ and $S_2^{hkl}$ can be assigned values picked from a handbook or computed using a simplified micromechanics model—such as the Reuss, Voigt, or Kröner model [9]. For accurate measurements, the X-ray elastic constants $S_1^{hkl}$ and $\frac{1}{2}S_2^{hkl}$ should be determined experimentally [10], for example, by following the procedure detailed in ASTM-E1426-14 [11]. The reason why standards recommend using experimentally determined values is that the X-ray elastic constants depend on the microstructure of the material and can vary from one specimen to the next, or even between regions of a specimen [12]. Another reason for using experimentally determined constants is that, since the crystallites located near the surface are likely to deform differently than crystallites located deeper into the material, different penetration depths will result in different X-ray elastic constants. This was recently pointed out in [13], which showed that this effect was all the more pronounced as the crystallites had anisotropic elastic properties and as their size was large compared to the maximum penetration depth of the X-rays.

In addition to being microstructure-dependent, several experimental studies, most of them published in the 1960s and 1970s, pointed out that the X-ray elastic constants can vary with plastic strain. In these studies, and in the present paper, the reported variations of XECs mean that the calibration parameter, needed to correlate an applied stress increment to the diffraction measurements via the $\sin^2\psi$ method, varies with plastic strain. For example, Marion and Cohen [8] reported that the X-ray elastic constants decreased by 40% in severely plastically deformed ARMCO-9 and 1045 steels as compared to model values. Esquivel [14] also observed variations in X-ray elastic constants with plastic deformation in various ferrous and non-ferrous alloys, and found that relative variations were larger for steels than for TA6V or aluminum alloys. Taira et al. [15,16] reported variations of X-ray elastic constants with plastic deformation in pure iron and in steels with a carbon content that ranged from 0.15%C to 0.91%C using Cr-K$\alpha$ and Co-K$\alpha$ radiations. The

researchers showed that X-ray elastic constant variations were larger when the carbon content increased, and explained this phenomenon by the stiffness difference between the ferrite and the carbon-rich cementite phase. Dölle et al. reported a decrease of the X-ray elastic constant in ferritic steels using {211} planes with Cr-K$\alpha$ radiation [17], and similar trends were found in copper using {331} planes with Cu-K$\alpha$ radiation [18].

Similar experiments conducted by Prümmer [19] on various steels have shown that, in addition to causing X-ray elastic constants to vary, plastic deformation also induced what is now called 'pseudo-macrostresses'. In an X-ray diffraction context, pseudo-macrostresses usually designate the average of the stress field over the diffracting domain. Because diffraction only samples a subset of all grains and phases, the average over the diffracting domain can differ from the true volume average and be wrongly interpreted as macrostresses. This can occur when observing only one phase of a multiphase material [20,21], or one set of diffracting planes in a plastically deformed material where intergranular stresses are present ([22,23] sec. 5.6). Prümmer showed that both the decrease in X-ray elastic constants and the amplitude of pseudo-macrostresses increased with carbon content, and that different incident radiations (and thus crystallographic planes) showed different sensibility to these phenomena.

Surprisingly, our literature review did not find any paper dealing with the variation in the X-ray elastic constants with plastic strain in the approximatively 40-year period that followed in the English-speaking literature. Only two publications reporting X-ray elastic constant evolution with plastic strain were found over the past 20 years [24,25].

Although the variation in the X-ray elastic constants with plastic deformation has been documented in all previously cited studies, there is, to the best of our knowledge, neither a clear explanation of the physical mechanisms responsible for this phenomenon nor an operational procedure to take it into account when measuring residual stresses using X-ray diffraction in parts that might be affected by it.

From Equation (2), the error on the measured stress is directly proportional to the error on the $\frac{1}{2}S_2^{hkl}$ constant. Therefore, the experimental calibration of the XEC should be performed every time quantitative results are expected. In practice, this is rarely done as it requires in situ testing devices and time, in order to measure the $\sin^2 \psi$ ellipse slope for various applied loads. Furthermore, even when the X-ray elastic constants are calibrated experimentally, calibration is seldom performed at more than one plastic strain level.

Yet, a wide variety of commonly used industrial processes that produce internal stresses also induce plastic strains such as cold-forming, welding, peening, machining, and turning [1]. For all parts and specimens produced using these processes, it is expected that the XEC will be affected. It implies that the XEC calibration should not be made for a single plastic strain state and that the stress calculation should account for the XEC corresponding to the plastic strain state in the gauge volume.

The objective of this study is to propose a method to calibrate X-ray elastic constants that considers the plastic strain that goes along the evaluated residual stress. For this purpose, the only available indication of plastic strain when performing X-ray measurements is the diffraction peak width. The developed methodology thus proposes to calibrate the XEC evolution with plastic strain and then to link the plastic strain to the peak width. It allows the XEC value to be finally linked to the measured peak width without the need for additional measurements. In this study, the method is applied to two different surface finishings of the same high-strength steel to highlight the importance of the constant calibration.

The paper is structured as follows. Section 2 describes the materials and methods. Section 3 presents the experimental results, focusing on the evolution of X-ray elastic constants with plastic deformation. It also introduces as an empirical strategy to take these variations into account when performing X-ray stress measurements. Finally, Section 4 discusses the findings and their limitations.

## 2. Materials and Methods

### 2.1. Material Composition, Microstructure, and Macroscopic Mechanical Properties (For Confidentiality Reasons, This Section Only Contains Limited Information)

Table 1 shows the nominal composition of the high-elastic limit ferritic-bainitic/perlitic steel used in this study. The material was sourced in the form of 40 mm thick cold-rolled plates. Figure 1 shows a typical micrograph and EBSD map acquired in the plane of the plate. Both reveal a microstructure with elongated (acicular) grains. According to the inverse pole figure shown in Figure 1b, the as-received material has a maximum texture index of 1.54, which indicates that it initially has almost no crystallographic texture.

**Table 1.** Elemental composition (mass%) of the investigated HLE steel.

| C | S | P | Ti | Mn | Ni | Cr | Mo | Fe |
|---|---|---|---|---|---|---|---|---|
| ≤0.150 | ≤0.005 | ≤0.015 | ≤0.250 | ≤0.550 | 4.5–5.0 | ≤0.550 | ≤0.5 | balance |

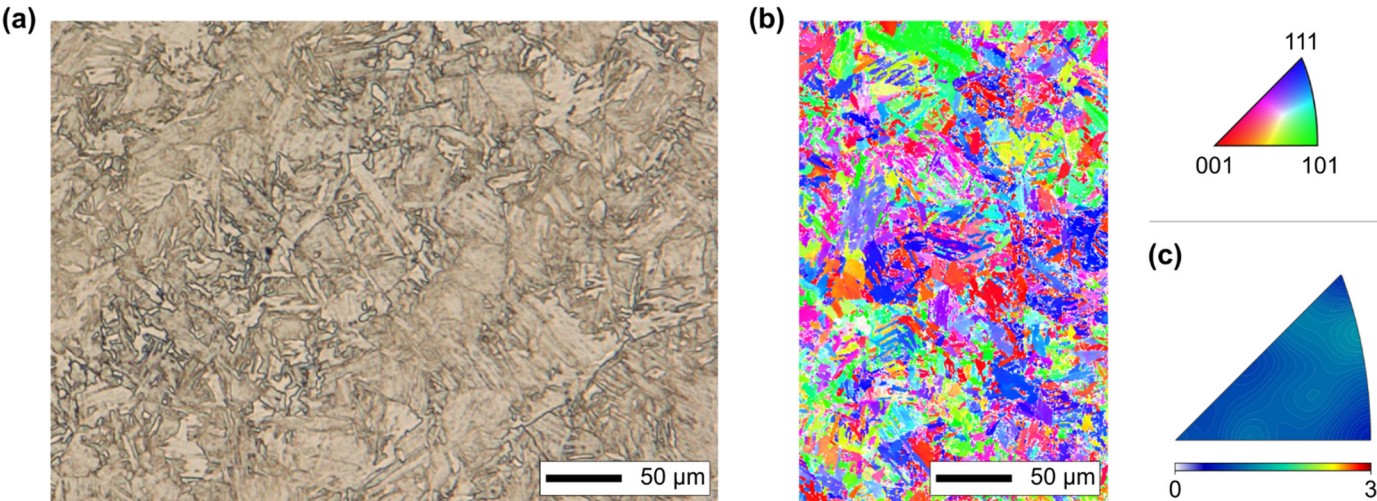

**Figure 1.** Microstructure of the high-elastic limit steel used in this study. (**a**) Typical optical micrograph in the plane of the plate after mirror polish and chemical attack with a 97% vol. ethanol, 3% vol. nitric acid solution. (**b**) Typical electron backscatter diffraction grain orientation map in the plane of the plate. (**c**) Inverse pole figure in the out-of-plane direction.

Conventional quasi-static tensile tests reported in [26] and conducted on specimens removed from the same plate found that Young's modulus is 205 GPa, Poisson's ratio is 0.3, and the yield stress at 0.2% is 670 MPa.

### 2.2. Test Specimens

Figure 2a shows the geometry of the specimen used for all measurements presented in the following sections. The specimens were removed from a 15 mm thick region located at the mid-thickness of the plate using wire electrical discharge machining, as shown in Figure 2b.

To remove initial residual stresses, the specimens were stress-relieved by heating them to 600 °C for two hours in a furnace filled with an inert argon atmosphere. They were cooled down in the furnace until the temperature fell below 150 °C, which took approximately 10 h, and then they were air-cooled to room temperature.

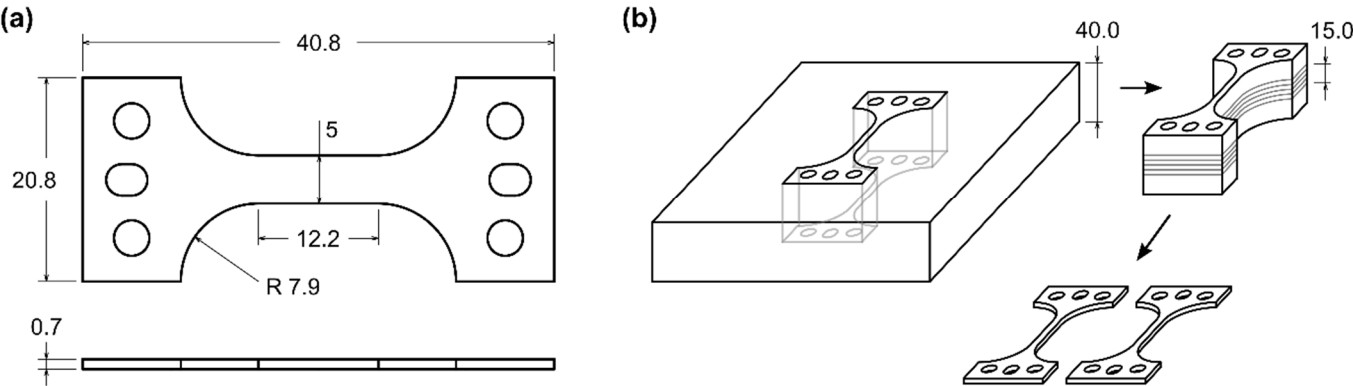

**Figure 2.** (**a**) Geometry of the tensile specimens used to determine X-ray elastic constants. X-ray stresses were measured at the center of the gauge length on one face while macroscopic strains were measured using a strain gauge attached to the same location on the opposite face. (**b**) Cutting plan. Specimens were removed using wire electrical discharge machining from a 15 mm thick region centered on the midplane of the plate, as illustrated schematically. All dimensions are in mm.

One specimen was put aside to be used in the as-annealed condition (i.e., straight out of the furnace), while three were electropolished to expose subsurface layers (Table 2). Electropolishing was performed using a Struers LectroPol machine (Struers, Copenhagen, Denmark) and Struer's A2 electrolyte. Electropolishing was performed symmetrically over the gauge length on both faces, turning over the specimen every 20 s until at least 70 μm were removed from each face.

**Table 2.** Lists of the specimens and conditions.

| Specimen ID | Condition |
| --- | --- |
| A1 | Annealed and electropolished |
| A2 | Annealed and electropolished |
| A3 | Annealed and electropolished |
| B1 | As-annealed |

*2.3. Procedure for Determining X-ray Elastic Constants*

2.3.1. Preliminaries

The general idea behind all procedures used to determine X-ray elastic constants (see, for example, [9] sec. 2.133a, [12,20,27], sec. 5.13) is to apply known macroscopic stresses to a specimen while staying well below the macroscopic yield stress, measuring lattice strains at each stress level by X-ray diffraction, and adjusting the $S_1^{hkl}$ and $\frac{1}{2} S_2^{hkl}$ constants until the model (Equation (2)) best fits the experimental data.

To minimize experimental error and ensure that measurements are not affected by initial residual stresses that might be present in the specimen, the ASTM-E1426-14 [11] and NF-EN-15305 [4] standards recommend to follow a variant of this procedure, shown schematically in Figure 3, which consists in applying a known uniaxial stress state $\sigma_{11}$ to a flat specimen (Figure 3a); measuring lattice strains at several tilt angles as in the standard $\sin^2 \psi$ method; determining the slope of the X-ray lattice strain vs. $\sin^2 \psi$ curves; repeating measurements for at least 5 stress levels (Figure 3b); plotting the previously determined slopes versus applied stresses; and determining the slope of the curve (Figure 3c). Then, provided that Equation (2) applies, the slope of the X-ray lattice strain vs. $\sin^2 \psi$ curves is equal to $\frac{1}{2} S_2^{hkl} \sigma_{11}$ and the slope of the last plot is $\frac{1}{2} S_2^{hkl}$ (refer to ASTM-E1426-14 for the detailed computation procedure).

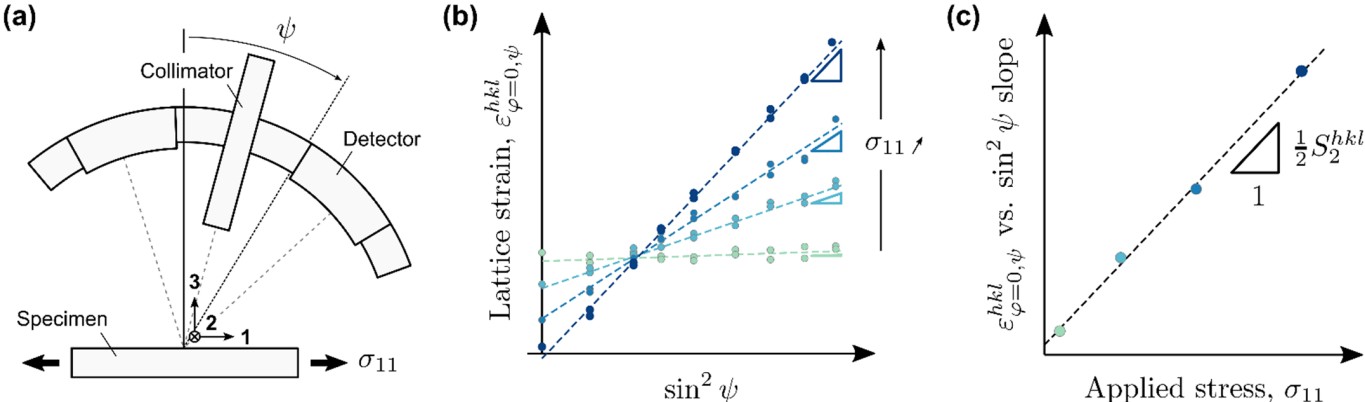

**Figure 3.** Schematic illustration of the experimental procedure used to determine X-ray elastic constants. (**a**) Typical experimental setup with a flat specimen loaded in uniaxial tension. Stresses are measured at the center of the specimen in the direction (11) in which the external load is applied. It should be noted that, for the sake of clarity, the figure shows a diffractometer in 'omega' mode, whereas measurements reported in this article were performed on a diffractometer in 'modified-chi' mode. (**b**) Lattice strains determined using X-ray diffraction versus $\sin^2 \psi$ for several values of applied stress. If the specimen has small grains compared to the size of the X-ray spot and has no texture, the points are expected to align, with a slope proportional to $1/2 S_2^{hkl} \sigma_{11}$. (**c**) Slopes of the dashed lines shown in (**b**) versus applied stress. The $1/2 S_2^{hkl}$ constant is determined as the slope of the best-fitting line.

2.3.2. Detailed Procedure

In this study, a Deben 5 kN micro-tensile stage was used to load the specimens. The applied loads were measured using the load cell of the tensile stage. The macroscopic strains were measured using a HBM 1-LY41-1.5-120 strain gauge attached to the back face of the specimen, directly below the X-ray spot.

Figure 4 illustrates the loading protocol. The specimen was first loaded up to the current maximum load with the crosshead of the tensile stage moving at 0.2 mm/s (1), then let to relax for 120 s (2). The specimen was then unloaded by 80 MPa to point A (Figure 4b). The X-ray elastic constants were determined as described in Section 2.3.1 using 9 stress levels linearly spaced on a 500 MPa range that started at point A. Measurements were performed at the 9 stress levels first for decreasing applied stress (3), then at the same 9 stress levels for increasing applied stress (4). Finally, the specimen was unloaded to measure residual X-ray stresses at zero applied stress (5).

This procedure was repeated for increasing maximum loads until the strain gauge failed. Measurements were automated and ran continuously one after the other. Each X-ray elastic constant measurement took approximately 2.5 h.

This procedure complies with the NF-EN-15305 standard (sec 10.1-5). It also follows most recommendations of the stricter ASTM-E1426-14 standard, except for the following three major deviations:

1.  Whereas ASTM-E1426-14 recommends using rectangular specimens with a length between grips at least four times the width and a width-to-thickness ratio less than eight, the tensile specimen's geometry described in Section 2.2 was used due to size and load constraints dictated by the micro-tensile stage used to load the specimen.

2.  Whereas ASTM-E1426-14 recommends measuring stresses experienced by the irradiated volume using calibrated strain gauges located as close as possible to the X-ray spot, the applied loads were measured using the micro-tensile stage load cell. This was because the steels' Young's modulus was known to decrease by several percentage points with plastic deformation [28]. Using strain gauges to measure stresses would have required re-calibrating the strain gauge at each plastic deformation level, which would have considerably lengthened measurements.

3. Whereas ASTM-E1426-14 recommends that applied stresses range from 10% to 70% of the macroscopic yield stress, a constant range of 500 MPa was sampled, with the higher applied stress equal to the macroscopic yield stress minus 80 MPa. This procedure was chosen to minimize micro-plasticity artifacts that appear at the end of the unloading on this material and that can be observed as the stress–strain curve forms an open loop.

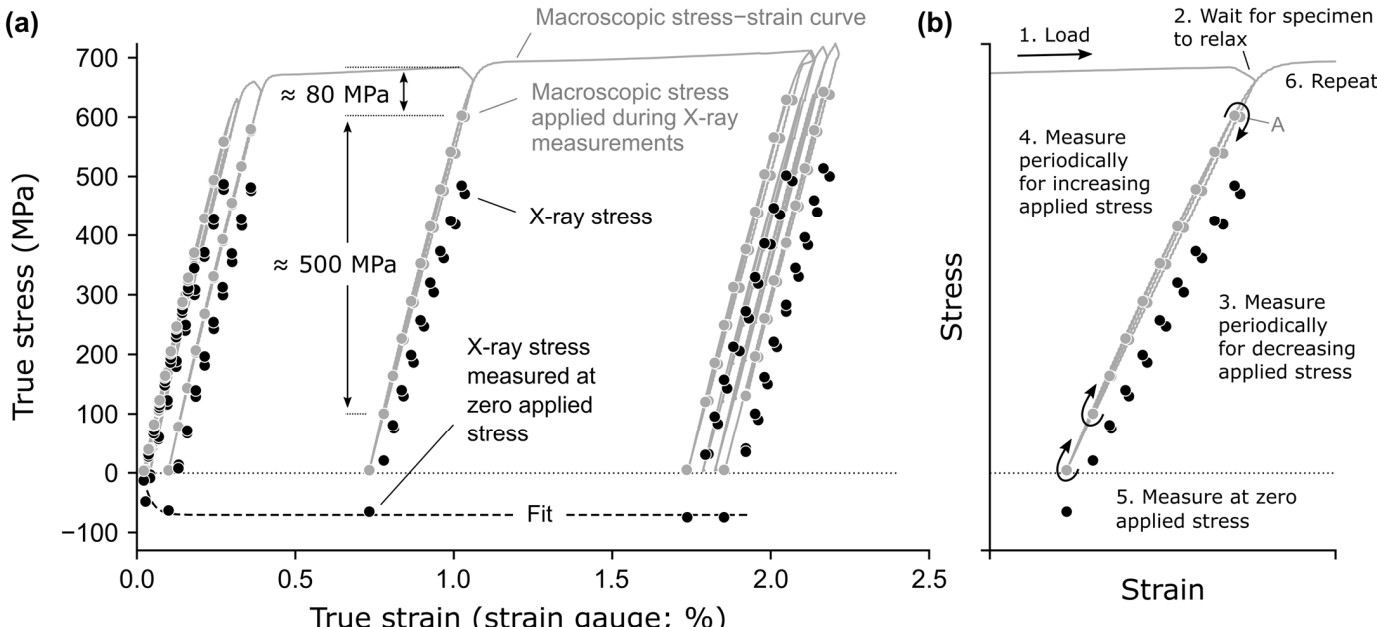

**Figure 4.** Macroscopic and X-ray stresses versus macroscopic strain. (**a**) Subset of stress–strain data acquired on specimen 1, where about half the dots were removed to avoid clutter. The grey line and dots show macroscopic stresses from the load cell; while black dots show stresses measured using X-ray diffraction. Each grey dot indicates where the loading was interrupted to perform X-ray stress measurement and is paired with one black dot. (**b**) Schematic illustration of the loading procedure. After loading and waiting for the specimen to relax, one X-ray stress measurement was performed at the locations marked by grey points. The specimen was then unloaded to measure residual X-ray stresses at zero applied stress.

*2.4. X-ray Stress Measurements*

2.4.1. Setup and Diffraction Conditions

All measurements were performed using a portable Stresstech G2R diffractometer (Stresstech, Jyväskylä, Finland) in modified-chi mode (NF EN 15305, [4]), equipped with two Dectris MYTHEN2 R 1D detectors (Dectris, Baden-Dättwil, Switzerland), a chromium anode, and a 2 mm diameter collimator.

Stresses were measured along the axis of the specimens only (i.e., at constant rotation angle) for the {211} planes of the ferrite phase ($2\theta \approx 156°$), using the $K_\alpha$ radiation of the chromium anode with the $\sin^2 \chi$ method—which is the analog of the standard $\sin^2 \psi$ method for diffractometers that operate in modified-chi mode; refer to appendix X1 of the ASTM-E2860-12 standard [5] for a detailed derivation of the model that relates lattice strains, goniometer angles, and components of the macroscopic stress tensor in macroscopically isotropic materials in modified-chi mode. Each measurement used 19 tilt angles linearly spaced between −41 and +41°, +/−3° tilt oscillations, and a 5 s exposure time. Vanadium $K_\beta$ filters were not used as none were available. The alignment of the diffractometer was checked according to ASTM-E915-96, using a calibration specimen made of iron powder embedded in an epoxy matrix. Figure 5 shows typical spectra acquired at 0° tilt (i.e., approximately normal to the surface of the specimen) on an electropolished (left) and as-annealed (right) specimen.

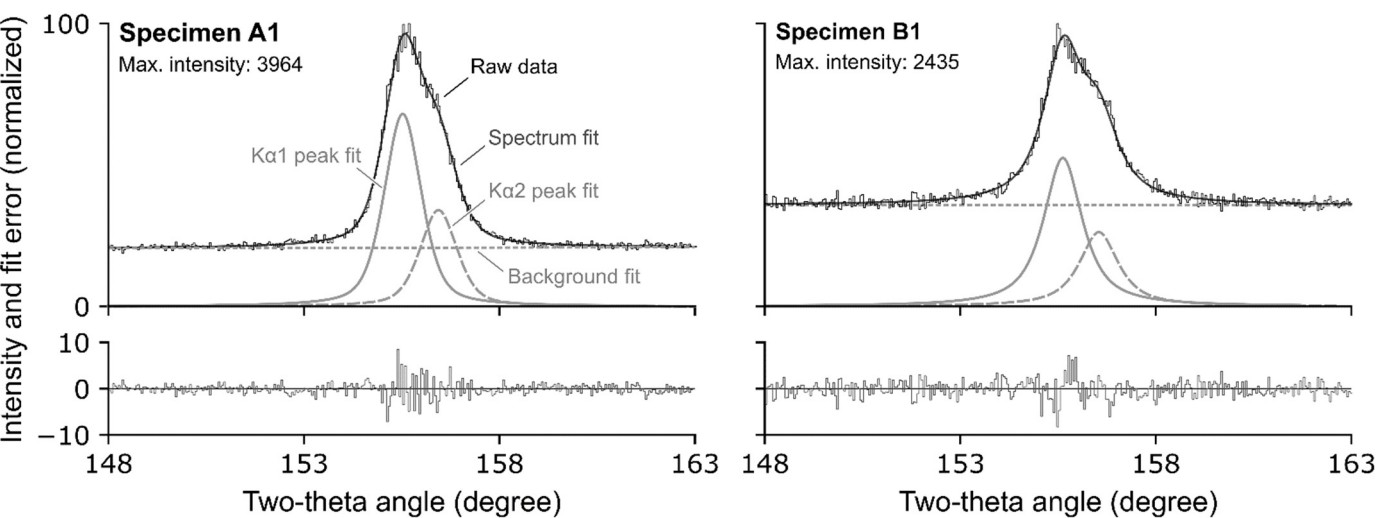

**Figure 5.** Typical raw diffraction spectra for an electropolished specimen (**left**) and as-annealed specimen (**right**). Both spectra were acquired on untested specimens at zero tilt—i.e., approximately normal to the surface—and normalized to make comparison easier. Also shown are the fitted spectra used to determine the location of the peaks, the components of the fit, and the fit error (bottom).

### 2.4.2. Peak Fitting

Diffraction spectra returned by the detectors, which had only undergone flat field correction, were post-processed using in-house codes. No Lorentz-polarization correction was applied as, unlike for omega and chi modes, this correction does not depend on the tilt angle in modified-chi mode. Each spectrum was fitted with a model consisting of the sum of a linear background term, a pseudo-Voigt function to represent the $K_{\alpha 1}$ peak, and a pseudo-Voigt function to represent the $K_{\alpha 2}$ peak, that is

$$f(x) = a_0 + a_1 x + f_{\text{pseudo}-\text{voigt}}(x|\alpha, \ p_0, \ p_1, p_2) + f_{\text{pseudo}-\text{voigt}}(x|\alpha^*, \ p_0^*, \ p_1^*, p_2^*), \quad (4)$$

where $a_0$, $a_1$, $\alpha$, $\alpha^*$, the $p_i$, and the $p_i^*$ are adjustable parameters,

$$\begin{aligned} f_{\text{pseudo}-\text{voigt}}(x|\ \alpha, \ p_0, p_1, p_2) \\ = (1-\alpha) \ f_{\text{gauss}}(x|\ p_0, p_1, p_2) + \alpha \ f_{\text{lorentz}}(x|\ p_0, p_1, p_2), \end{aligned} \quad (5)$$

$$f_{\text{gauss}}(x, |p_0, p_1, p_2) = p_0 \exp\left[-\ln(2)\frac{(x-p_1)^2}{p_2^2}\right], \quad (6)$$

and

$$f_{lorentz}(x|p_0, p_1, p_2) = \frac{p_0}{1 + \frac{(x-p_1)^2}{p_2^2}}. \quad (7)$$

In the above expressions, $x$ is the $2\theta$ angle, $a_0$ and $a_1$ describe the linear background, $p_0$ and $p_0^*$ the height of the peaks, $p_1$ and $p_1^*$ their center, $p_2$ and $p_2^*$ their half width at half maximum, and $\alpha$ and $\alpha^*$ are weights that vary between 0 and 1.

Because both $K_{\alpha 1}$ and $K_{\alpha 2}$ peaks were generated by the same diffractometer, it is assumed that both peaks had the same shape, that is $\alpha^* = \alpha$ and $p_2^* = p_2$. Furthermore, since the intensity of the $K_{\alpha 2}$ peak is known to be half that of the $K_{\alpha 1}$ peak ([22], sec. 1.4), the condition $p_0^* = p_0/2$ was imposed. Finally, because the wavelengths of the radiations that cause each peak are constants for a given anode, writing Braggs law for each radiation and dividing one by the other yields

$$\frac{\sin \theta_2}{\sin \theta_1} = \frac{\lambda_2}{\lambda_1} \Rightarrow \theta_2 = \arcsin\left(\sin \theta_1 \frac{\lambda_2}{\lambda_1}\right), \quad (8)$$

where $\lambda_i$ is the wavelength of the radiation that causes the $K_{\alpha i}$ peak and $\theta_i$ is the associated Bragg angle (between the incident and diffracted beam). This implies that

$$\frac{p_1^*}{2} = \arcsin\left(\sin\left(\frac{p_1}{2}\right)\frac{\lambda_2}{\lambda_1}\right). \tag{9}$$

Put together, the relationships from this paragraph enable to deduce all parameters that describe the $K_{\alpha 2}$ peak from those that describe the $K_{\alpha 1}$ peak, which reduces the number of parameters to optimize from 10 to 6.

Fitting was performed using Scipy's nonlinear least-squares solver, least_squares. The gradient of the cost function was computed analytically. Initial estimations of the parameters were computed from the experimental spectrums as follows: $p_0$ as the maximum intensity of the spectrum for the height of the peak; $p_1$ as the location of the maximum intensity; $p_2$ as the area divided by twice the maximum height and 0.5 for $\alpha$. Figure 5 shows typical results of the fitting procedure.

### 2.4.3. X-ray Stress Computations

For stress computations, lattice spacing was computed using Bragg's law as

$$d_{\phi,\psi}^{hkl} = \frac{1}{2}\frac{\lambda_{K_{\alpha 1}}}{\sin\theta_{K_{\alpha 1}}},$$

where $\lambda_{K_{\alpha 1}}$ is the $K_{\alpha 1}$ wavelength, and $\theta_{K_{\alpha 1}}$ is the diffraction angle of the $K_{\alpha 1}$ peak; lattice strains were computed as

$$\epsilon_{\phi,\psi}^{hkl} = \frac{d_{\phi,\psi}^{hkl} - d_0}{d_0},$$

where the reference lattice spacing $d_0 = 0.117089$ nm was measured at zero tilt on a stress-free powder. Finally, macroscopic stresses were computed using the $\sin^2\chi$ method (the analog of the standard $\sin^2\psi$ method for diffractometers in modified-$\chi$ configuration) as described in standards NF EN 15305 and ASTM-E2860-12.

## 3. Results

Figure 4 illustrates the raw experiment performed on specimen A1. The grey line is the true stress–strain curve recorded by the load cell and strain gauge. The black dots are the XRD measurements performed at different strain levels during load and unload phases. For these measurements, the $\frac{1}{2}S_2^{hkl}$ constant was set to the textbook value of $5.76 \times 10^{-6}$ MPa$^{-1}$ [5]. In an ideal scenario, the black dots would superimpose with the stress–strain curve. However, this is not the case due to the two phenomena already mentioned in the introduction.

The first phenomenon is due to pseudo-macrostresses that causes stresses measured using X-ray diffraction to differ from zero at zero applied stress. As in the previously cited studies, compressive pseudo-macrostresses were measured, the amplitude of which increases with plastic strain. This offset stress evolution is represented with a black dashed line in Figure 4 and plotted for four tested specimens in Figure 6. The evolution of pseudo-macrostresses depends on the surface condition. For electropolished specimens, pseudo macrostresses are initially close to zero and reach $-50$ to $-75$ MPa at 1.8% of added plastic strain. For as-annealed specimens, the pseudo-macrostresses are initially compressive, at about $-20$ MPa, then increase up to 20 MPa, before falling back to almost zero.

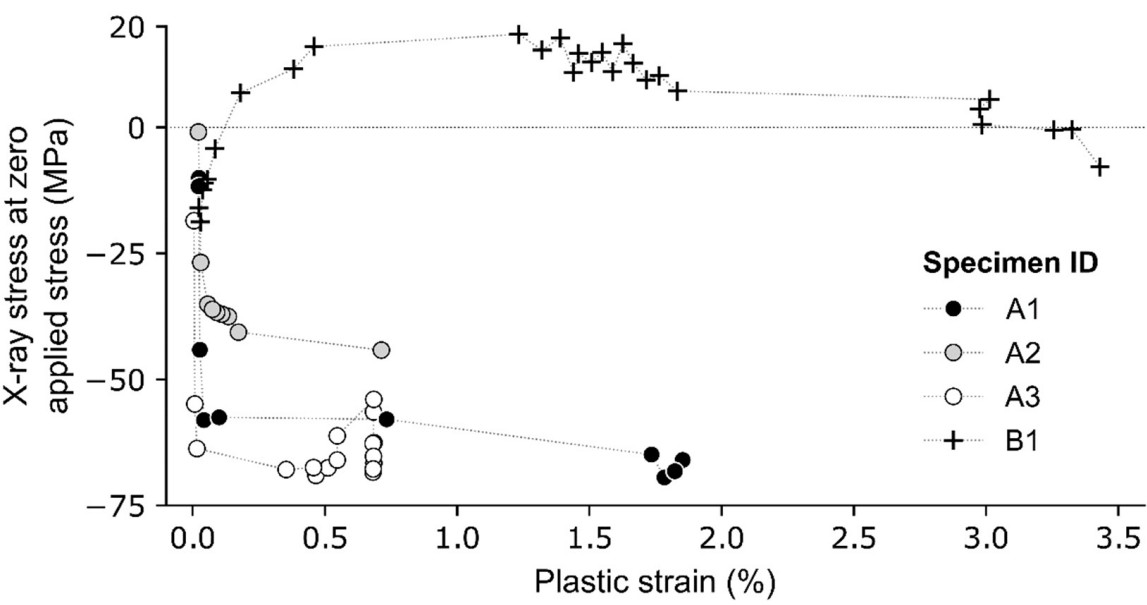

**Figure 6.** X-ray stresses at zero applied stress versus plastic strain for the four tested specimens.

The second phenomenon is the variation of X-ray elastic constants with plastic strain. This is illustrated in Figure 7 which shows the evolution of experimentally determined $\frac{1}{2}S_2^{hkl}$ versus plastic strain. As for pseudo-macrostresses, the shape of the curves depends on the surface condition.

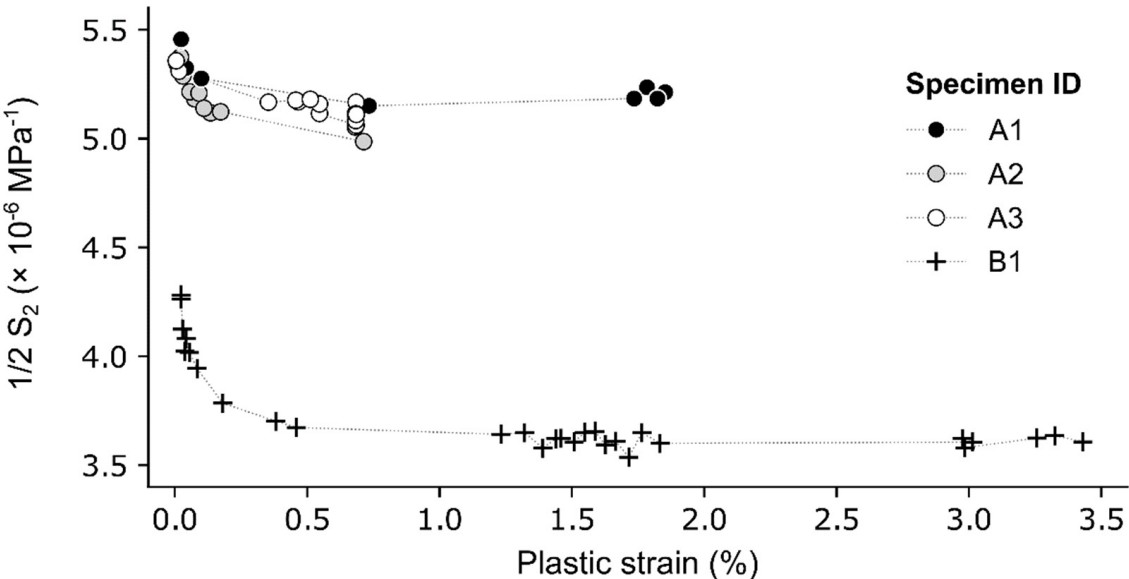

**Figure 7.** X-ray elastic constant $\frac{1}{2}S_2^{hkl}$ versus plastic strain for the four tested specimens.

-   For electropolished specimens, the constant is initially close to values reported in the literature, and then decreases by about 6% after 2% added plastic strain.
-   For as-annealed specimens, the constant is initially about 25% lower than the values from the literature, at about $4.3 \times 10^{-6}$ MPa$^{-1}$, and it further decreases by about 13%, down to $3.75 \times 10^{-6}$ MPa$^{-1}$.

This study aims at proposing a methodology to account for this second issue in order to calculate stresses with the adequate $\frac{1}{2}S_2^{hkl}$ value. However, the plastic strain in the gauge volume is not a generally available piece of information when performing diffraction measurements. The diffraction peak width is known to be an indirect measurement of

the plasticity level. Figure 8 presents the evolution of the integral breadth as a function of plastic strain for the four tested samples. Values vary with plastic strain from 1.5° to 2° for electropolished samples and from 1.9° to 2.7° for the as-annealed specimen.

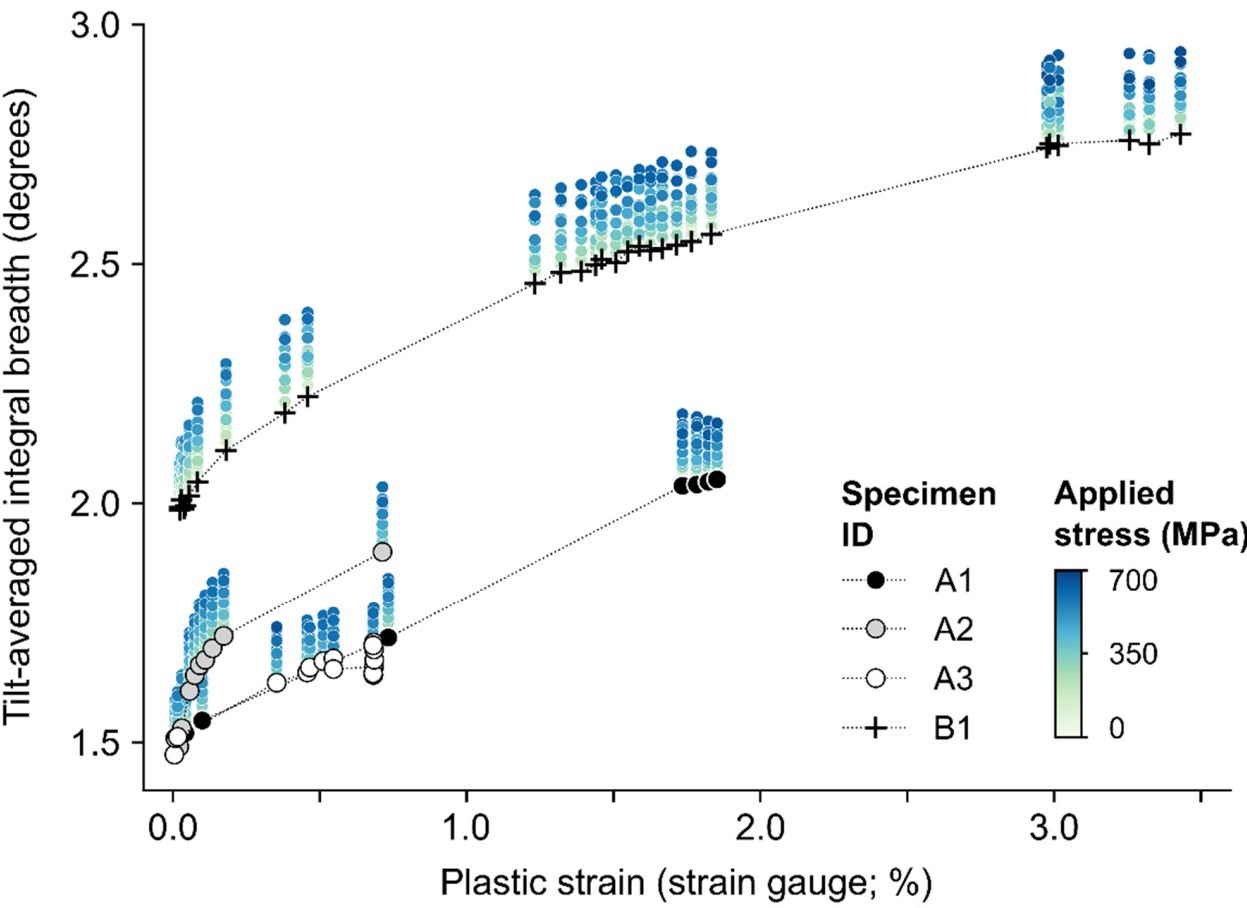

**Figure 8.** Tilt-averaged integral breadth versus macroscopic plastic strains. Black and white markers and the dotted curve highlight values at zero applied stress, whereas colored dots show how tilt-averaged integral breadth increases with applied stress.

These values are not only linked to the material and its plastic strain state but also to the XRD measurement device, and therefore only the relative variation with plastic strain of the integral breadth between the samples of this study can be looked at. Absolute values cannot be compared to values in the literature as for the XEC.

## 4. Discussion

As integral breadth values can be linked to plastic strain and $\frac{1}{2}S_2$ values decrease with plastic strain, it is possible to evaluate the $\frac{1}{2}S_2^{hkl}$ parameter as a function of the integral breadth. The evolution is plotted in Figure 9 and exhibits a non-linear behavior, followed by an asymptotic stabilized behavior. For this purpose, a decreasing exponential fit function is chosen.

$$\frac{1}{2}S_2(B) = \frac{1}{2}S_2^{ini} - \left(\frac{1}{2}S_2^{ini} - \frac{1}{2}S_2^{final}\right) \times (1 - \exp(-\xi(B - B_{ini}))) \qquad (10)$$

where $\xi$ is the only parameter that needs to be optimized on the set of data (here, using Scipy's non-linear least-squares optimization algorithm). The other parameters are directly calculated as the average on each group of experiments (electropolished and as-annealed) as follows:

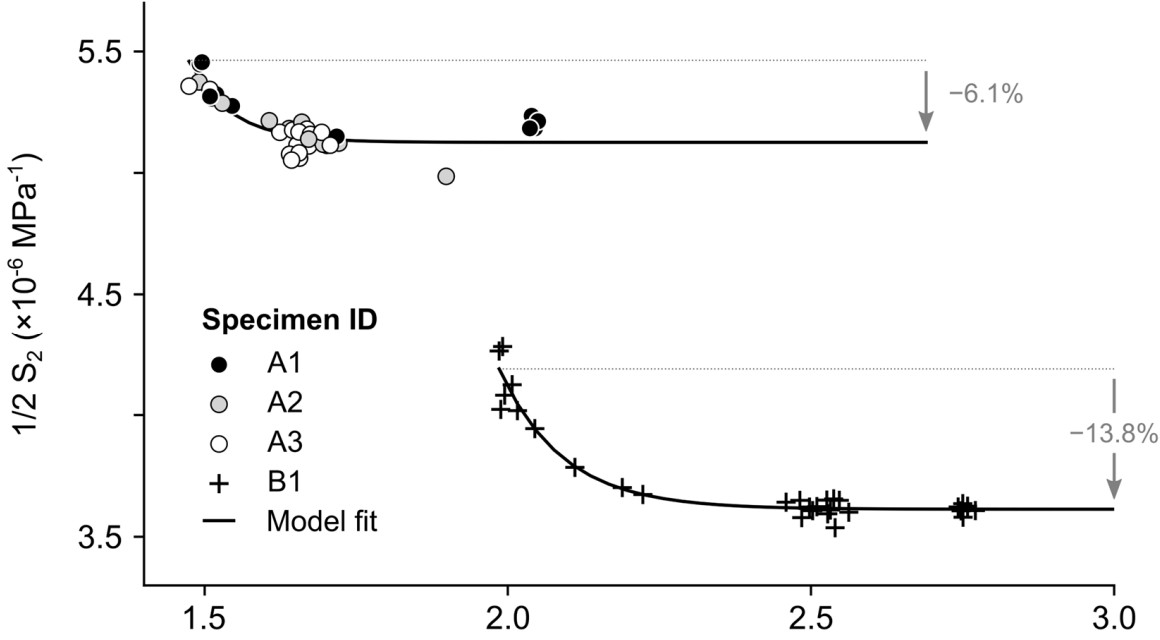

**Figure 9.** Experimentally determined $\frac{1}{2}S_2^{hkl}$ versus tilt-averaged integral breadth at zero applied stress and exponential decay fits. Points for specimens 1 to 3 were fitted together.

- $\frac{1}{2}S_2^{ini}$ is the $\frac{1}{2}S_2^{hkl}$ value assessed on the first load of the test;
- $\frac{1}{2}S_2^{final}$ is the $\frac{1}{2}S_2^{hkl}$ value assessed on the last unload of the test;
- $B_{ini}$ is the tilt-averaged integral breadth at the start of the test.

Table 3 shows the fitted parameters and Figure 9 shows the fitted model superimposed to experimental data.

**Table 3.** Fitting parameters obtained for the two sets of data.

| Surface Condition/Parameter | $1/2S_2^{ini}$ ($\times 10^{-6}$ MPa$^{-1}$) | $1/2S_2^{final}$ ($\times 10^{-6}$ MPa$^{-1}$) | $B_{ini}$ (Degree) | $\xi$ (Degree$^{-1}$) |
|---|---|---|---|---|
| Electropolished | 5.459 | 5.126 | 1.475 | 14.33 |
| As-annealed | 4.189 | 3.612 | 1.986 | 9.52 |

Using these evolutive $\frac{1}{2}S_2^{hkl}$ parameters, the XRD measured stress values can be corrected. To visualize the effect of this correction without results being polluted by pseudo-macrostresses, Figure 10 shows increments of X-ray stress—i.e., X-ray stress minus X-ray stress at zero applied stress—of all stress measurements plotted versus applied stress. Each set of points differs by the $\frac{1}{2}S_2$ parameters used for computing X-ray stresses:

- Black dots use the constant textbook value $\frac{1}{2}S_2^{hkl} = 5.76 \times 10^{-6}$ MPa$^{-1}$ and represent the case where no calibration test is performed (i.e., the same value for all specimens).
- Grey dots represent the standard case where a calibration test is performed on each specimen, and stresses are calculated using the experimentally determined $\frac{1}{2}S_2$ value obtained at the beginning of each test (i.e., a different value for each specimen).
- White dots are obtained with the methodology proposed in this study: $\frac{1}{2}S_2^{hkl}$ values are computed with Equation (10), using the parameters listed in Table 3 (i.e., a different value for each specimen and each plastic strain level).

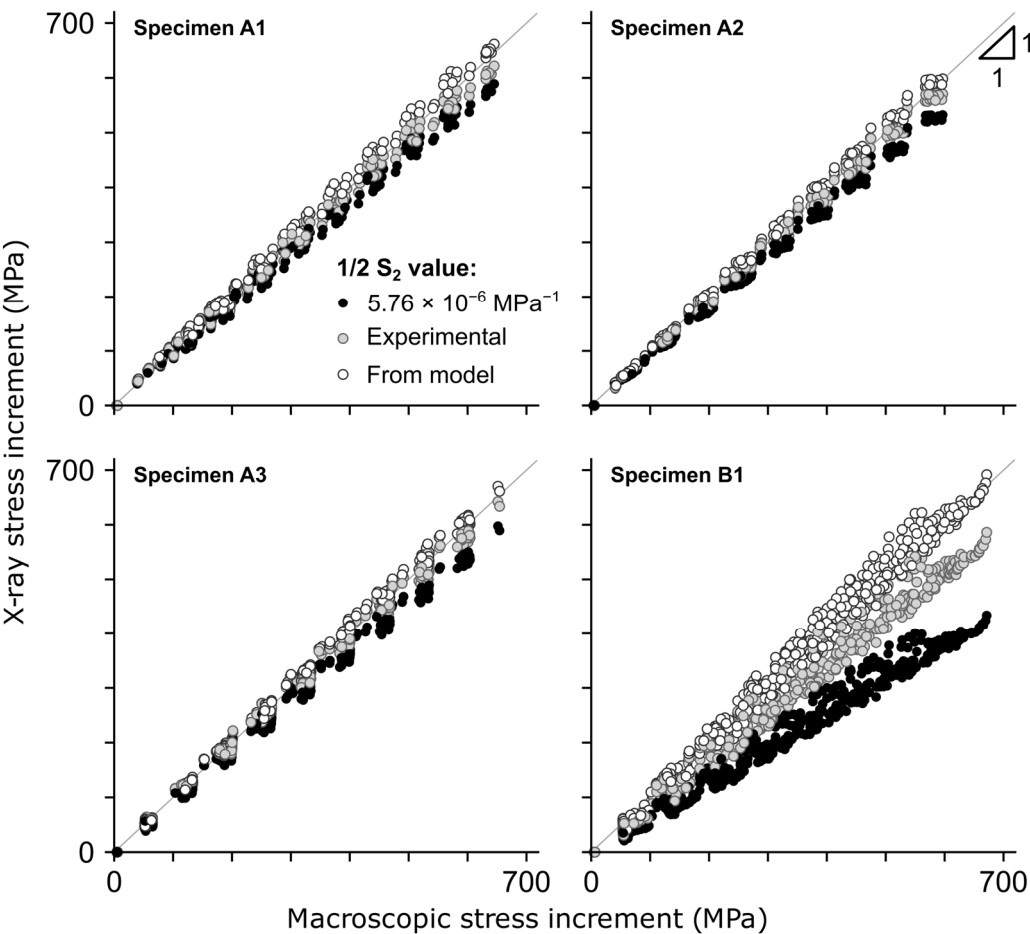

**Figure 10.** Scatter plots showing experimentally determined X-ray stress increments—i.e., X-ray stress minus X-ray stress at zero applied stress—versus applied stress. Black dots show X-ray stresses computed with a constant textbook-value of $\frac{1}{2}S_2$, grey dots show X-ray stresses computed with experimentally determined $\frac{1}{2}S_2$ (Figure 7), and white dots show X-ray stresses computed using the model in Figure 9. In a hypothetical perfect scenario, all points would converge along an unitary slope line.

It is clear that the proposed methodology is the option where the points best correspond to the unitary slope line and thus the proposed methodology is efficient to enhance XRD-measured stress accuracy. The importance of the XEC correction is emphasized on the as-annealed sample where its variation is the most important.

As the proposed methodology is time-consuming (60 h/specimen) and requires automation to enhance measurement accuracy and repeatability, a simplified calibration can be made using only the initial condition of the specimen, without adding plastic strain to the specimen, as recommend by the standards. This was performed for the calculation of the grey dots. Although it degrades results, compared to the proposed methodology of this paper, it still significantly enhances the accuracy of the measurements compared to the black dots calculated with the handbook value. This is due to the fact that, even with thermally stress-relieved specimens, the calibrated $\frac{1}{2}S_2^{hkl}$ values are significantly lower than the handbook value, respectively −5% for the electropolished specimens and −25% for the as-annealed specimen.

The observations in Figure 10 are similar to the conclusion that is drawn from Figure 7 which validates two important points:

-   Whether it is a constant or variable parameter with plastic strain, the effect of taking into account the calibrated $\frac{1}{2}S_2^{hkl}$ values is more important than other stress measurement uncertainties.

- This study is based on the experimental observation that the $\frac{1}{2}S_2^{hkl}$ parameter depends on plastic strain. In order to propose a practical solution, the peak width is used as a variable instead of plastic strain to describe the XEC evolution. However, it induces a secondary dependence of the XEC on the stress level as the peak width depends on both plastic strain and stress levels (see Figure 8). Figure 10 proves that the methodology does not suffer from this bias for the studied configurations.

The effect of the correction on the absolute stress value accuracy also depends on the pseudo-macrostress level. For electropolished specimens, the contribution of the variation of $\frac{1}{2}S_2^{hkl}$ is small ($\approx$5%), compared to the contribution of pseudo-macrostress that can be as high as $-70$ MPa for plastic strains of a few tenths of a percent. In other words, the error introduced by neglecting pseudo-macrostresses is significantly larger than that introduced by neglecting the variation of X-ray elastic constants with plastic strain. For this surface condition, the relevance of the proposed correction method is therefore limited, especially if the amplitude of residual stresses is small. On the other hand, for as-annealed specimens, the contribution of the variation of $\frac{1}{2}S_2^{hkl}$ is more important ($\approx$15% decrease) whereas the contribution of pseudo-macrostresses is smaller ($\pm$20 MPa).

In some rare cases, uncertainty on the XEC is not a problem and the proposed methodology is not necessary. For example, when post-processing treatment parameters are optimized based on the optimal observed fatigue life of a material, it can be assumed that the uncertainty is the same during the parameter's optimization phase and the subsequent production control tests, as long as the measurement protocol is constant.

In most cases, however, XEC uncertainty is a major issue for the following stress evaluation purposes:

- Stress values can be used to validate process simulation models. In this case, also, absolute macroscopic stress values are of interest and constants should be corrected.
- More critically, stresses can be used for design purposes. The mean stress effect determines fatigue life [29,30], corrosion cracking [31,32], and fracture failure [33,34]. The corresponding design criteria are generally calibrated on simple specimens designed for laboratory testing. In the case where the surface and plastic strain states of the calibration specimen differ from those of the application cases, different constants should be used for the stress evaluation of the two configurations, which is not generally done as it is assumed that the material is the same.

To conclude on this part, it must be reminded that if the calibrated XEC depends on the plastic deformation, it also depends strongly of the surface condition. Common practice is to consider that XEC depends on the diffracting plane and material microstructure. However, the surface condition also makes a major contribution to the uncertainty of the XEC value, which leads to direct uncertainty on the assessed stress values.

## 5. Conclusions

This study presented a series of in situ X-ray stress measurements performed on small specimens of a high-elastic limit steel loaded in uniaxial tension. The experimental results reveal that the $\frac{1}{2}S_2^{hkl}$ X-ray elastic constant decreases with plastic strain, which is consistent with earlier studies from the literature conducted on other metals.

Although the variation of $\frac{1}{2}S_2^{hkl}$ with plastic strain is documented since the 1950s, to the best of our knowledge, no method has been proposed to take this effect into account when performing X-ray stress measurements on plastically deformed parts. In fact, although international standards recommend using experimentally determined X-ray elastic constants, from our experience, these measurements are seldom conducted in industrial practice, with most measurements using tabulated X-ray elastic constant values.

The main contribution of this work is to propose a methodology to correct XRD measurements using a $\frac{1}{2}S_2^{hkl}$ value calibrated to the tilt-averaged integral breadth. For this purpose, a calibration function is used. This procedure allows the method to be transposed to any sample at no additional cost as once the function parameters are known, all information needed is obtained during a classic XRD measurement.

The proposed methodology can be applied to any material suited to X-ray stress measurements, assuming a one-to-one relation between peak width and $\frac{1}{2}S_2^{hkl}$, which implies that the influence of the stress level on peak width is neglected. The application of the method on a thermally stress-relieved high-strength steel submitted to uniaxial tensile testing with two surface preparations has proven the efficiency of the method in this particular case. The methodology remains to be validated for other materials, as well as for other loading paths.

Apart from the $\frac{1}{2}S_2^{hkl}$ variation, this study also highlights the fact that the X-ray elastic constant can vary strongly with the near-surface microstructure. Measurements on EDM-cut and annealed specimens have shown that the variation of the X-ray elastic constant is very sensitive to the surface state of the specimen as these samples exhibit a relatively important X-ray elastic constant decrease ($-14\%$) compared to the same samples which underwent additional electropolishing ($-6\%$ variation).

It is also important to note that the initial X-ray elastic constant value of the electropolished specimen of about $5.42 \times 10^{-6}$ MPa$^{-1}$ was close to the $5.76 \times 10^{-6}$ MPa$^{-1}$ textbook value, whereas the initial value for the as-annealed specimen of about $4.28 \times 10^{-6}$ MPa$^{-1}$ is lower. This highlights the importance of calibrating the X-ray elastic constant value on the exact same surface state, whether this value is assumed to be constant or not with respect to the plastic strain. The total obtained X-ray elastic constant values vary from 3.5 to $5.5 \times 10^{-6}$ MPa$^{-1}$, leading to proportional errors on the evaluated stress.

Further investigations would include testing additional surface states on the same material, such as laser or shot-peened surfaces, as well as testing the influence of multiaxial loading paths on the X-ray elastic constant evolution.

**Author Contributions:** Software, P.F.; investigation, E.G. and P.F.; writing—original draft preparation, E.G., B.L. and P.F.; writing—review and editing, L.B., S.C. and C.D.; supervision, B.L., L.B., S.C. and C.D. All authors have read and agreed to the published version of the manuscript.

**Funding:** This research received no external funding.

**Data Availability Statement:** The data presented in this study are available on request from the corresponding author. The data are not publicly available due to privacy.

**Acknowledgments:** The authors acknowledge support from the Institut Carnot Arts through the I-DEF-X project supplemented by IRDL and MSMP self-fundings. The authors would like to thank The French Armed Forces Ministry and the Defense Innovation Agency (AID) for the financial support of this work. The authors would like to thank the Brittany region, the FEDER, and the Finistère Department Council for the funding of the diffractometer and tensile testing system used in this work, through the CPER IF-SYS-MER.

**Conflicts of Interest:** The authors declare no conflicts of interest.

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
