# Peer review of "A Simple Calibration Method to Consider Plastic Deformation Influence on X-ray Elastic Constant Based on Peak Width Variation"

_metals, doi:10.3390/met14010062_

Round 1
Reviewer 1 Report
Comments and Suggestions for Authors
This is a meaningful work, which is well organized and written. I just found a minor error in line 46 which needs to be revised.
Comments on the Quality of English Languageit's good.
Author Response
Thanks for your careful reading, the error has been corrected
Reviewer 2 Report
Comments and Suggestions for Authors
This is an interesting, well written and easy to read article. Can the authors however, consider the following issues?
11) In line 43 there is figure number missing. Please make necessary correction.
22) According to Fig. 1a, the microstructure of the experimental steel consists mainly of needles, plates and laths of, presumably, martensite, bainite and Widmansttäten ferrite. How did you define and measure an average grain of of 11 mm?
33) In Fig. 1, does the 50 mm bar applies to both the micrograph and the orientation map? It would be better to include a micron bar for the micrograph.
44) To obtain a lattice strain from measuring of lattice spacings a reference lattice spacing is needed. What was the reference d-spacing used?
Reviewer 3 Report
Comments and Suggestions for Authors
The work proposes a procedure for calibrating the X-Ray parameter used for residual stress measurement in polycrystalline materials.
The scientific relevance of the work appears to be limited; however, the method may be useful in the application field and, for that reason, may be considered for publication.
Nevertheless, some minor issues need to be resolved.
Line 46 refers to a figure "xxx" that is not present.
In Figure 4 several critical issues are present. The location of point "A" does not seem consistent with what is stated in the text. Similarly, the unloading of 80 MPa is incorrectly represented.
Also, keeping a constant value of 80 MPa for the first unloading, the analyzed stress range varies with plastic deformation due to work hardening. This disagrees with what is stated in item 3 (lines 244-248) and should be clarified.
Author Response
The work proposes a procedure for calibrating the X-Ray parameter used for residual stress measurement in polycrystalline materials.
The scientific relevance of the work appears to be limited; however, the method may be useful in the application field and, for that reason, may be considered for publication.
Nevertheless, some minor issues need to be resolved.
Line 46 refers to a figure "xxx" that is not present.
Thank you for your careful reading, the error has been corrected
In Figure 4 several critical issues are present. The location of point "A" does not seem consistent with what is stated in the text. Similarly, the unloading of 80 MPa is incorrectly represented.
Thank you for your careful reading again. There was an error in the figure due to the different versions. The figure has been modified
Also, keeping a constant value of 80 MPa for the first unloading, the analyzed stress range varies with plastic deformation due to work hardening. This disagrees with what is stated in item 3 (lines 244-248) and should be clarified.
The sentence that started in line 248 (now 255) was modified :
This procedure was chosen to minimize micro-plasticity artifacts that appear at the end of the unloading on this material and that can be observed as the strain-stress curve forms an open loop.